# SIMPLER CALIBRATION FOR SURVIVAL ANALYSIS

## ABSTRACT

Survival analysis, also known as time-to-event analysis, is the problem to predict the distribution of the time of the occurrence of an event. This problem has applications in various fields such as healthcare, security, and finance. While there have been many neural network models proposed for survival analysis, none of them are calibrated. This means that the average of the predicted distribution is different from the actual distribution in the dataset. Therefore, X-CAL has recently been proposed for the calibration, which is supposed to be used as a regularization term in the loss function of a neural network. X-CAL is formulated on the basis of the widely used definition of calibration for distribution regression. In this work, we propose new calibration definitions for distribution regression and survival analysis, and demonstrate a simpler alternative to X-CAL based on the new calibration definition for survival analysis.

## 1 INTRODUCTION

*Survival analysis*, also known as *time-to-event analysis*, is the problem to predict the time of the occurrence of an event. In healthcare applications, the event typically corresponds to a death or the onset of disease in a patient. The time between a well-defined starting point and the occurrence of the event is called the *survival time* or *failure time*. In survival analysis, we usually estimate the *distribution* of the survival times of patients. Survival analysis has important applications in healthcare as well as various other fields (e.g., credit scoring (Dirick et al., 2017) and fraud detection (Zheng et al., 2019)). The recent progress of prediction models for survival analysis has been summarized in a survey paper (Wang et al., 2019).

In survival analysis, datasets are often *censored*, which means that events of interest might not be observed for some instances. This may be due to either the limited observation time window or missing traces caused by other irrelevant events. Typical censored data are right censored data. These are the data points whose exact times of the events are unknown; we know only that the events had not happened up to a certain time. In this paper, we focus on the *uncensored* data and the *right censored* data, as shown in Figure 1. Here, the event for data point $x_1$ is observed during the period of study and hence this data is categorized as uncensored data. The data points $x_2$ and $x_3$ are categorized as right censored data because we did not observe the events during the period of study. The time between a well-defined starting point and the last observation time (e.g., the time of the end of study) is called the *censoring time*.

One of the classical methods to solve the survival analysis problem is the Kaplan-Meier estimator (Kaplan & Meier, 1958). This is a non-parametric method to estimate the distribution of the survival times as a survival function $S(t)$, where the value $S(t^*)$ for a specific time $t^*$ represents the *survival rate* at time $t^*$ (i.e., the ratio of the patients who survived at time $t^*$). It is easy to estimate the survival function $S(t)$ if the dataset contains only uncensored data points, but the Kaplan-Meier estimator is designed to work for datasets that include censored data. Here, we briefly explain the algorithm of the Kaplan-Meier estimator. Let $\{t_i\}_{i=1}^k$ be the set of distinct times when at least one uncensored event was observed in the dataset. Let $d_i$ be the number of (uncensored) events that happened exactly at time $t_i$, and let $n_i$ be the number of data points that are known to have survived at time $t_i$. Then, the Kaplan-Meier estimator outputs the survival function

$$S(t) = \prod_{i:t_i \leq t} \left(1 - \frac{d_i}{n_i}\right).$$

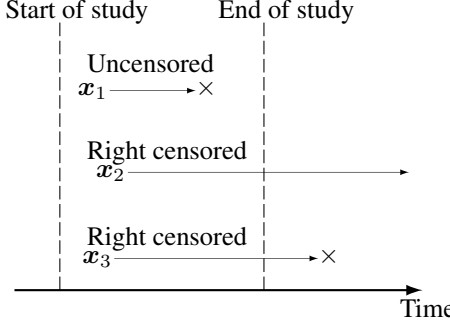
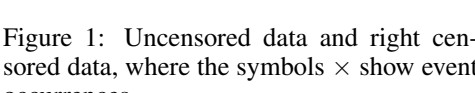

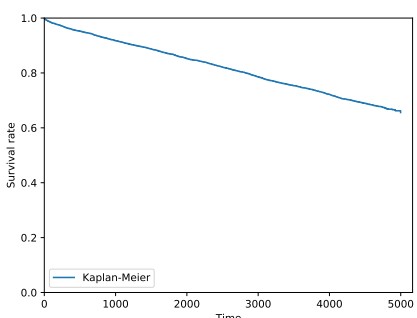

Figure 1: Uncensored data and right censored data, where the symbols $\times$ show event occurrences.

Figure 2: Survival function $S(t)$ estimated by the Kaplan-Meier estimator for the flchain dataset.

Figure 2 shows an example of the survival function $S(t)$ estimated by the Kaplan-Meier estimator for the flchain dataset (Dispenzieri et al., 2012). Here, we can see that the survival rate at time $t = 2500$ is approximately 80%. Note that the true survival rate $S(t)$ at $t = \infty$ must be zero, but the Kaplan-Meier estimator outputs the survival function $S(t)$ only for time $t \in [0, t_{\max}]$, where $t_{\max}$ is the maximum survival time of the uncensored data points in a dataset. This is because we cannot estimate the survival rate $S(t)$ for the time $t > t_{\max}$.

A drawback of the Kaplan-Meier estimator is that it outputs a survival function $S(t)$ for the entire population and not for a specific patient. Therefore, there have been many algorithms to estimate the survival rate $S(t|x)$ for each patient $x$ so as to enable personalized medicine (Wang et al., 2019). In particular, many neural network models that predict the survival function $S(t|x)$ have been proposed (Lee et al., 2018; Ren et al., 2019; Zheng et al., 2019; Tjandra et al., 2021).

## 1.1 CALIBRATION

When we use a prediction model, it should be *calibrated*. In binary classification, this means that a prediction model that outputs a confidence of a true label for an input is expected to satisfy the condition that the average of the confidence values over all inputs are equal to the ratio of the data points with a true label in the dataset. For example, if a dataset contains 40% of the data points with a true label, then we expect that a calibrated prediction model will output a confidence of 0.4 on average. Even though calibration is important for prediction models, and neural network models have been widely used for prediction models, (Guo et al., 2017) showed that neural network models are often miscalibrated.

In regression analysis, quantile-based calibration is widely used as the definition of calibration (Kuleshov et al., 2018; Song et al., 2019; Cui et al., 2020; Zhao et al., 2020). In survival analysis, the definition of calibration is based on this definition for regression analysis. Goldstein et al. (2020) showed a method to train a neural network model to achieve calibration by adding a new regularizer, X-CAL, to the loss function of the neural network to achieve calibration during the training. This method is in contrast to the widely-used calibration methods for regression analysis such as Platt scaling and isotonic regression, which are post-training methods.

Note that a *calibrated* model is not necessarily useful. For example, in binary classification we can construct a trivial calibrated model that always outputs the ratio of true label data points in a dataset as a confidence value. Therefore, a useful prediction model also needs to be *sharp*, which means that it outputs a confidence value close to one or zero for each input. To obtain a prediction model that is both calibrated and sharp, one approach is to train a neural network with a loss function consisting of one loss term aimed at sharp prediction and another aimed at calibration. X-CAL utilizes this approach in that it is a regularizer for calibration and is used in combination with another loss term that aims at sharp prediction.

## 1.2 OUR CONTRIBUTIONS

In this paper, we propose a *Kaplan-Meier regularizer* as a simpler alternative to X-CAL. An advantage of our regularizer is that the obtained prediction model is calibrated for any time $t \in [0, t_{\max}]$. Another advantage is that our regularizer does not require any hyperparameter, whereas X-CAL requires hyperparameters. The idea behind our new regularizer is simple: just to reduce the difference between the average predicted survival function and the Kaplan-Meier survival function (see Figure 3). Our new regularizer is based on a new definition of calibration for survival analysis, and we discuss its advantages in Section 4.

## 2 SURVIVAL ANALYSIS

We formally describe the problem settings of regression analysis and survival analysis. In *regression analysis*, we assume that there is an unknown probability distribution $\mathcal{P}$ on the sample space $\mathcal{X} \times \mathcal{Y}$, where $\mathcal{X}$ is the domain of features and $\mathcal{Y}$ is the interval $[-\infty, \infty]$. We refer to the associated random variables with capital letters (i.e., $\boldsymbol{X}$ and $\boldsymbol{Y}$) and realizations with lower case letters $(\boldsymbol{x}, y) \sim \mathcal{P}$. We assume that we can obtain independent samples $\{(\boldsymbol{x}_i, y_i)\}_{i=1}^n$ from $\mathcal{X} \times \mathcal{Y}$ according to distribution $\mathcal{P}$.

In *survival analysis*, we assume that there is an unknown probability distribution $\mathcal{Q}$ on the sample space $\mathcal{X} \times \mathcal{T} \times \mathcal{C}$, where $\mathcal{X}$ is the domain of features and $\mathcal{T}$ and $\mathcal{C}$ are time intervals $[0, \infty]$. We refer to the associated random variables with capital letters (i.e., $\boldsymbol{X}, \boldsymbol{T}$, and $\boldsymbol{C}$) and realizations with lower case letters $(\boldsymbol{x}, t, c) \sim \mathcal{Q}$. The random variable $\boldsymbol{T}$ corresponds to the survival time (i.e., the time of the occurrence of an event), which might not be observable due to the censoring, and the random variable $\boldsymbol{C}$ corresponds to the censoring time. We assume that the censoring time is $\infty$ for an uncensored event. The random variable defined by $\boldsymbol{Z} = \min\{\boldsymbol{T}, \boldsymbol{C}\}$ corresponds to the time of the last observation (i.e., the survival time or the censoring time). Different from the regression analysis, we cannot obtain samples $\{(\boldsymbol{x}_i, t_i, c_i)\}_{i=1}^n$ from $\mathcal{X} \times \mathcal{T} \times \mathcal{C}$ according to distribution $\mathcal{Q}$ due to the censoring in survival analysis. However, we assume that we can obtain independent samples $D = \{(\boldsymbol{x}_i, z_i, \delta_i)\}_{i=1}^n$ instead, where $\delta_i$ is a binary value indicating if the $i$-th data point is censored or not. Here an uncensored data point $(\boldsymbol{x}_i, z_i, \delta_i) \in D$ satisfies $\delta_i = 1$ and $z_i = t_i$ and a censored data point $(\boldsymbol{x}_i, z_i, \delta_i) \in D$ satisfies $\delta_i = 0$ and $z_i = c_i$. Hence $\delta_i = 0$ means that we know only the fact that $t_i \geq c_i$ and the exact survival time $t_i$ is unknown.

The task of survival analysis is to predict the probability of an event of interest occurring at time $t$ for $\boldsymbol{x} \in \mathcal{X}$ as a probability distribution function $f(t|\boldsymbol{x})$. This function $f(t|\boldsymbol{x})$ is often represented in other equivalent forms. For example, it can be represented as its cumulative distribution function (CDF)

$$F(t|\boldsymbol{x}) = \int_0^t f(\tau|\boldsymbol{x})d\tau$$

or as the survival function $S(t|\boldsymbol{x})$ defined by

$$S(t|\boldsymbol{x}) = 1 - F(t|\boldsymbol{x}).$$

Intuitively, $F(t|\boldsymbol{x})$ represents the probability of observing the event by time $t$ for $\boldsymbol{x}$ and the survival function $S(t|\boldsymbol{x})$ represents the probability of not-observing the event until time $t$ for $\boldsymbol{x}$.

Many neural network models have been proposed for survival analysis (e.g., (Lee et al., 2018; Ren et al., 2019; Zheng et al., 2019; Tjandra et al., 2021)). They output the probability distribution in the form of $f_\theta(t|\boldsymbol{x})$, $F_\theta(t|\boldsymbol{x})$, $S_\theta(T|\boldsymbol{x})$, or something equivalent to these forms, where $\theta$ is the parameters of the neural network.

## 3 QUANTILE-BASED CALIBRATION

In this section, we review the definitions of *calibration* for distribution regression and survival analysis in the literature. First, we consider the *distribution regression* whose task is to predict the distribution of the target variable as a CDF $F_\theta(y|\boldsymbol{x})$ for $\boldsymbol{x} \in \mathcal{X}$, where $\theta$ is the parameters of the prediction model. In distribution regression, quantile-based calibration (Kuleshov et al., 2018) is widely used as the definition of calibration (e.g., (Song et al., 2019; Cui et al., 2020; Zhao et al., 2020)).

**Definition 3.1** *(Quantile-based calibration.) A prediction model $F_\theta(y|\boldsymbol{x})$ for distribution regression is* quantile-calibrated *if this equation holds for any quantile level $\tau \in [0, 1]$:*

$$\Pr_{(\boldsymbol{X},\boldsymbol{Y})\sim\mathcal{P}}(F_\theta(\boldsymbol{Y}|\boldsymbol{X}) \leq \tau) = \tau. \tag{1}$$

If we can compute the inverse of $F_\theta(y|\boldsymbol{x})$, we can rewrite Eq. (1) as

$$\Pr_{(\boldsymbol{X},\boldsymbol{Y})\sim\mathcal{P}}(\boldsymbol{Y} \leq F_\theta^{-1}(\tau|\boldsymbol{X})) = \tau.$$

This equation means that, for a random sample $(\boldsymbol{x}, y) \sim \mathcal{P}$, $y$ must be at most the $\tau$-th quantile of the predicted CDF (i.e., $F_\theta^{-1}(\tau|\boldsymbol{x})$) exactly with probability $\tau$. We can rewrite Definition 3.1 in another equivalent formulation: the following equation holds for any subinterval $[\tau_1, \tau_2] \subseteq [0, 1]$:

$$\Pr_{(\boldsymbol{X},\boldsymbol{Y})\sim\mathcal{P}}(F_\theta(\boldsymbol{Y}|\boldsymbol{X}) \in [\tau_1, \tau_2]) = \tau_2 - \tau_1.$$

This equation means that the quantile level $F_\theta(y|\boldsymbol{x})$ predicted for a random sample $(\boldsymbol{x}, y) \sim \mathcal{P}$ is contained in a subinterval $[\tau_1, \tau_2] \subseteq [0, 1]$ exactly with probability $\tau_2 - \tau_1$.

On the basis of Definition 3.1, Goldstein et al. (2020) define calibration for *survival analysis*.

**Definition 3.2** *(Quantile-based calibration for survival analysis.) A prediction model $F_\theta(t|\boldsymbol{x})$ for survival analysis is* quantile-calibrated *if this equation holds for any quantile level $\tau \in [0, 1]$:*

$$\Pr_{(\boldsymbol{X},\boldsymbol{T},\boldsymbol{C})\sim\mathcal{Q}}(F_\theta(\boldsymbol{T}|\boldsymbol{X}) \leq \tau) = \tau. \tag{2}$$

We can rewrite Definition 3.2 into another equivalent formulation: the following equation holds for any subinterval $I = [\tau_1, \tau_2] \subseteq [0, 1]$:

$$\Pr_{(\boldsymbol{X},\boldsymbol{T},\boldsymbol{C})\sim\mathcal{Q}}(F_\theta(\boldsymbol{T}|\boldsymbol{X}) \in I) = |I| = \tau_2 - \tau_1. \tag{3}$$

A problem when using Definition 3.2 is that we cannot get samples directly from $\boldsymbol{T}$ due to the censoring. As such, we cannot verify Eq. (3) for datasets that include censored data. Goldstein et al. (2020) resolved this problem by showing how to estimate the probability $\Pr(F_\theta(t|\boldsymbol{x}) \in I)$ for any $t \in [c, \infty]$ from $\Pr(F_\theta(c|\boldsymbol{x}) \in I)$ for a randomly sampled *censored* data point $(\boldsymbol{x}, c, 0)$. Under the assumption that $\boldsymbol{T}$ and $\boldsymbol{C}$ are independent (i.e., $\boldsymbol{T} \perp \boldsymbol{C} \mid \boldsymbol{X}$), they show

$$\Pr(F_\theta(t|\boldsymbol{x}) \in I) = \frac{(\tau_2 - v)\mathbb{1}[v \in I]}{1 - v} + \frac{(\tau_2 - \tau_1)\mathbb{1}[v < \tau_1]}{1 - v}, \tag{4}$$

where $v = F_\theta(c|\boldsymbol{x})$ and $\mathbb{1}[\cdot]$ denotes the step function. By using this estimation, we can compute the left-hand side of Eq. (3) from dataset $D$ that include censored data points.

On the basis of Eq. (3) and the approaches described in (Andres et al., 2018; Haider et al., 2020), Goldstein et al. (2020) proposed a metric called *distributional calibration* (D-CAL), which is defined as

$$\ell_{\mathrm{D-CAL}}(\theta) = \sum_{I \in \mathcal{I}} \left( \mathbb{E}_{(\boldsymbol{X},\boldsymbol{T},\boldsymbol{C})\sim\mathcal{Q}}\mathbb{1}[F_\theta(\boldsymbol{T}|\boldsymbol{X}) \in I] - |I| \right)^2,$$

where the collection $\mathcal{I}$ is chosen to contain disjoint contiguous subintervals of $C \subseteq [0, 1]$ that cover the whole interval $[0, 1]$.

A prediction model $F_\theta(t|\boldsymbol{x})$ with a lower $\ell_{\mathrm{D-CAL}}(\theta)$ is said to be more calibrated, but we cannot construct a neural network model that directly minimizes D-CAL because $\ell_{\mathrm{D-CAL}}(\theta)$ is not a differentiable function due to its step function. Therefore, Goldstein et al. (2020) defined the *explicit calibration* (X-CAL), an approximation of D-CAL, by replacing the step function with a sigmoid function so that X-CAL becomes a differentiable function. Moreover, X-CAL is designed to handle a set of data points $B = \{(\boldsymbol{x}_i, z_i, \delta_i)\}_{i=1}^{b}$ as a mini-batch, which makes it possible to integrate X-CAL into the loss function of neural network models because most of those models use a mini-batch training rather than the full batch training. Formally, X-CAL is defined as

$$\mathcal{R}_{\mathrm{X-CAL}}(\theta) = \mathbb{E}_{B\sim\mathcal{Q}} \sum_{I \in \mathcal{I}} \left( \mathbb{E}_{(\boldsymbol{X},\boldsymbol{T},\boldsymbol{C})\sim B}\zeta(F_\theta(\boldsymbol{T}|\boldsymbol{X}); I, \gamma) - |I| \right)^2,$$

where $\zeta(z; I, \gamma)$ is a sigmoid function to approximate the step function in D-CAL. (See (Goldstein et al., 2020) for the precise definition of the function $\zeta$.) Here, we abuse notation $(\boldsymbol{X}, \boldsymbol{T}, \boldsymbol{C}) \sim B$ to indicate that we obtain sample data points from mini-batch $B$ (rather than the probability distribution $\mathcal{Q}$). Note that Goldstein et al. (2020) proposed using X-CAL in the loss function of a neural network model as a regularizer, which means that it is intended to be combined with other loss functions. This is because a prediction model that aims only at calibration is useless (as discussed in Section 1) and the balance between sharpness and calibration must be considered for obtaining a useful prediction model.

## 4 VALUE-BASED CALIBRATION

In this section, we propose alternative definitions of calibration, *value-based calibration*, for distribution regression and survival analysis. Then, on the basis of the new calibration definition for survival analysis, we propose our new *Kaplan-Meier regularizer*, and we discuss its advantages over D-CAL and X-CAL.

We first propose an alternative definition of calibration for distribution regression.

**Definition 4.1** *(Value-based calibration.) A prediction model $F_\theta(y|\boldsymbol{x})$ for distribution regression is* value-calibrated *if this equation holds for any value $y \in [-\infty, \infty]$:*

$$\mathbb{E}_{(\boldsymbol{X},\boldsymbol{Y})\sim\mathcal{P}}F_\theta(y|\boldsymbol{X}) = \Pr_{(\boldsymbol{X},\boldsymbol{Y})\sim\mathcal{P}}(\boldsymbol{Y} \leq y).$$

This equation means that the average probability of prediction having a value of at most $y$ must be equal to the actual ratio of data points having a value of at most $y$. The difference between Definitions 3.1 and 4.1 is on the choice of the axis. Whereas Definition 3.1 gives a natural condition for calibration with respect to the $\tau$-axis of a prediction model $\tau = F_\theta(y|\boldsymbol{x})$, Definition 4.1 gives a natural condition for calibration with respect to the $y$-axis. In appendix (Section A.1), we show that a quantile-calibrated model is not necessarily a value-calibrated model and vice versa.

On the basis of Definition 4.1, we define the value-based calibration for survival analysis.

**Definition 4.2** *(Value-based calibration for survival analysis.) A prediction model $F_\theta(t|\boldsymbol{x})$ for survival analysis is* value-calibrated *if this equation holds for any time $t \in [0, t_{\max}]$, where $t_{\max}$ is the maximum time of the uncensored data points in the dataset $D$:*

$$\mathbb{E}_{(\boldsymbol{X},\boldsymbol{T},\boldsymbol{C})\sim\mathcal{Q}}F_\theta(t|\boldsymbol{X}) = \Pr_{(\boldsymbol{X},\boldsymbol{T},\boldsymbol{C})\sim\mathcal{Q}}(\boldsymbol{T} \leq t). \tag{5}$$

Note that we ask Eq. (5) to hold for $t \in [0, t_{\max}]$ rather than $t \in [0, \infty]$. This is because the prediction model $F_\theta(t|\boldsymbol{x})$ for survival analysis is usually trained for $t \in [0, t_{\max}]$ and the prediction $F_\theta(t|\boldsymbol{x})$ for $t > t_{\max}$ is less accurate.

We can rewrite Definition 4.2 into another equivalent form: the following equation holds for any subinterval $[t_1, t_2] \subseteq [0, t_{\max}]$:

$$\mathbb{E}_{(\boldsymbol{X},\boldsymbol{T},\boldsymbol{C})\sim\mathcal{Q}}(F_\theta(t_2|\boldsymbol{X}) - F_\theta(t_1|\boldsymbol{X})) = \Pr_{(\boldsymbol{X},\boldsymbol{T},\boldsymbol{C})\sim\mathcal{Q}}(t_1 \leq \boldsymbol{T} \leq t_2).$$

This condition exactly matches the explanation of calibration in (Goldstein et al., 2020), i.e., a model's predicted number of events within any time interval is similar to the observed number. Note also that we can rewrite Eq. (5) into another equivalent form:

$$\mathbb{E}_{(\boldsymbol{X},\boldsymbol{T},\boldsymbol{C})\sim\mathcal{Q}}S_\theta(t|\boldsymbol{X}) = \Pr_{(\boldsymbol{X},\boldsymbol{T},\boldsymbol{C})\sim\mathcal{Q}}(\boldsymbol{T} > t). \tag{6}$$

Here, the left-hand side of Eq. (6) is changed from the CDF $F_\theta(t|\boldsymbol{X})$ in Eq. (5) to the survival function $S_\theta(t|\boldsymbol{X})$, and the right-hand side of the equation is also changed accordingly.

On the basis of Eq. (6), we propose our new metric, *Kaplan-Meier loss*, which is defined as

$$\ell_{\mathrm{KM}}(\theta) = \int_0^{t_{\max}} \left(\mathbb{E}_{(\boldsymbol{X},\boldsymbol{T},\boldsymbol{C})\sim\mathcal{Q}}S_\theta(t|\boldsymbol{X}) - \Pr_{(\boldsymbol{X},\boldsymbol{T},\boldsymbol{C})\sim\mathcal{Q}}(\boldsymbol{T} > t)\right)^2 dt. \tag{7}$$

In contrast to D-CAL, we can use this loss directly in the loss function of a neural network model because this loss is differentiable. However, we also propose a new regularizer $\mathcal{R}_{\mathrm{KM}}(\theta)$ by modifying $\ell_{\mathrm{KM}}(\theta)$ so that it can be used in a mini-batch training with a set of data points $B = \{(\boldsymbol{x}_i, z_i, \delta_i)\}_{i=1}^b$:

$$\mathcal{R}_{\mathrm{KM}}(\theta) = \mathbb{E}_{B\sim\mathcal{Q}}\int_0^{t_{\max}} \left(\mathbb{E}_{(\boldsymbol{X},\boldsymbol{T},\boldsymbol{C})\sim B}S_\theta(t|\boldsymbol{X}) - \Pr_{(\boldsymbol{X},\boldsymbol{T},\boldsymbol{C})\sim B}(\boldsymbol{T} > t)\right)^2 dt. \tag{8}$$

Again, we abuse notation $(\boldsymbol{X}, \boldsymbol{T}, \boldsymbol{C}) \sim B$ to indicate that we obtain sample data points from mini-batch $B$ (rather than the probability distribution $\mathcal{Q}$). Although we used $\ell_2$ loss in Eq. (7)–(8), we can use any other metric to measure the difference between two distributions. Note that we can compute the first term in the parentheses by

$$\mathbb{E}_{(\boldsymbol{X},\boldsymbol{T},\boldsymbol{C})\sim B}S_\theta(t|\boldsymbol{X}) = \frac{1}{|B|} \sum_{(x_i,z_i,\delta_i)\in B} (1 - F_\theta(t|x_i)),$$

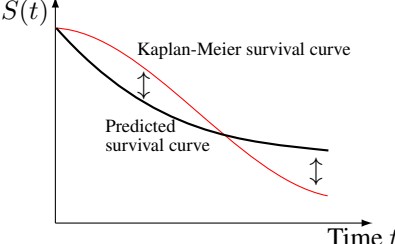

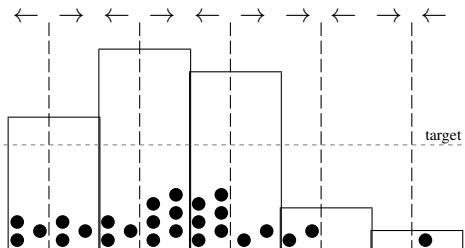

Figure 3: Our Kaplan-Meier regularizer is designed to reduce the $\ell_2$ loss between the Kaplan-Meier survival curve and the average predicted survival curve.

Figure 4: Each box corresponds to a bin in $\mathcal{C}$ and each circle corresponds to a data point. The arrows show the directions of the moves of the data points during training.

and we can estimate the second term in the parentheses $\Pr_{(\boldsymbol{X},\boldsymbol{T},\boldsymbol{C})\sim\mathcal{Q}}(\boldsymbol{T} > t)$ by the Kaplan-Meier estimator. Recall that the Kaplan-Meier estimator is designed to estimate $\Pr(\boldsymbol{T} > t)$ even if the dataset contains censored data. Figure 3 illustrates the proposed loss $\ell_{\mathrm{KM}}(\theta)$ and regularizer $\mathcal{R}_{\mathrm{KM}}(\theta)$, in which the first term in the parentheses of Eq. (7)–(8) corresponds to the average predicted survival function (thick curve) and the second term corresponds to the Kaplan-Meier survival curve (red curve) and each equation computes the $\ell_2$ loss between these two curves.

The advantages of our approach can be summarized as follows.

1. Our value-based calibration for survival analysis (Definition 4.2) is intuitive for practitioners in healthcare. Practitioners are often interested in the prediction performance at a specific time $t^* \in [0, t_{\max}]$, and Eq. (5) shows a natural condition that should hold at time $t^*$. In contrast, Eq. (2) states nothing about specific time $t^*$.

2. Our value-based calibration for survival analysis (Definition 4.2) is defined only for the range $t \in [0, t_{\max}]$. Regarding the quantile-based calibration, Definition 3.1 means that, if a prediction model $F_\theta(y|\boldsymbol{x})$ for distribution regression is quantile-calibrated, then the distribution of $F_\theta(\boldsymbol{Y}|\boldsymbol{X})$ for $(\boldsymbol{X}, \boldsymbol{Y}) \sim \mathcal{P}$ is equal to the uniform distribution over $[0, 1]$ (Zhao et al., 2020). However, even if a prediction model $F_\theta(t|\boldsymbol{x})$ for survival analysis is quantile-calibrated, the distribution of $F_\theta(\boldsymbol{T}'|\boldsymbol{X})$ is usually not equal to the uniform distribution when $\boldsymbol{T}'$ refers to the random variable for the survival time of *uncensored* events. This means that the condition in Definition 3.2 can be satisfied only if we can estimate the distribution of $F_\theta(\boldsymbol{T}|\boldsymbol{X})$ from the dataset $D$ that include censored data by using some method such as Eq. (4). In other words, we need to estimate $F_\theta(t|\boldsymbol{x})$ for $t > t_{\max}$ when we use Definition 3.2.

3. Our Kaplan-Meier loss and regularizer do not require any hyperparameter (other than the batch size $|B|$), whereas D-CAL and X-CAL require the hyperparameters $\gamma$ and $\mathcal{C}$.

4. Our Kaplan-Meier regularizer uses a simple $\ell_2$ loss and it avoids the binning approach used in X-CAL. In the following, we explain the problem in the binning approach. In X-CAL, we count the number of data points in each bin and we update the parameters $\theta$ of a neural network during the training so that the numbers of data points in the bins are balanced. Figure 4 illustrates that, for a bin in which the number of data points exceed the target, the parameters $\theta$ is updated so that the data points in the first half of the bin are pushed to the left and the data points in the second half of the bin are pushed to the right due to the approximation of the step function as the sigmoid function. However, since the numbers of data points in the three bins in the left exceed the target and the numbers of the data points in the two bins in the right are below the target in the case of Figure 4, no data point in the three bins in the left should be pushed left and the data points in these bins should be used to fill the two bins in the right. Therefore, we sometimes find difficulties in minimizing X-CAL due to the binning approach. Note that Goldstein et al. (2020) showed how to avoid this phenomenon for the left-most and right-most bins, but we cannot avoid this problem for the other bins.

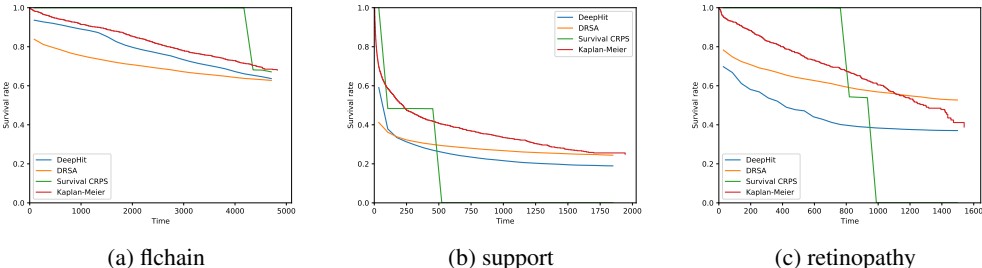

|              |              |                |
|:------------:|:------------:|:--------------:|
| (a) flchain  | (b) support  | (c) retinopathy |

Figure 5: Comparison of average predicted survival functions by using loss functions DeepHit, DRSA, and Survival CRPS with Kaplan-Meier survival curve.

## 5  EXPERIMENTS

In this section, we compare the performance of our Kaplan-Meier regularizer with X-CAL on real datasets. We show that, by using our Kaplan-Meier regularizer or X-CAL, we can reduce both Kaplan-Meier loss and D-CAL. This means that we can use one of these two regularizers both for quantile-based calibration and value-based calibration and do not need to combine the two regularizers for two different definitions of calibration.

### 5.1  NEURAL NETWORK AND DATASETS

We constructed a single neural network for our experiments and combined it with various loss functions. This neural network was a two-layer perceptron with a single hidden layer containing 128 neurons, and the number of outputs was 32. The activation function after the hidden layer was the ReLU type, and the activation function at the output node was softmax. We used Python 3.7.4 and PyTorch 1.4.0 for the implementation. The Adam optimizer (Kingma & Ba, 2015) was utilized for the training algorithm, with the learning rate set to 0.001, the batch size to 1024, and the other parameters to their default values. We run training for 100 epochs.

We used three datasets for survival analysis: two obtained from the packages in R (R Core Team, 2016) and one from a private data source. Of these former two, one is the flchain dataset (Dispenzieri et al., 2012), which was obtained from the 'survival' package and contains 7874 data points (69.9% of which are censored), and the other is the support dataset (Knaus et al., 1995), which was obtained from the 'casebase' package and contains 9104 data points (31.9% of which are censored). The dataset from the private data source is of patients with retinopathy disease. This dataset contains 6951 data points (33.3% of which are censored).

We split each of the three datasets into the training (80%) and test (20%) sets by using random partitioning. The neural network model was trained using the training set and the experimental results shown in this section (including the Kaplan-Meier survival curves) were obtained using the test set. When we computed the metrics (D-CAL and Kaplan-Meier loss) on a test dataset, we used full batch (rather than the mini-batch). Regarding the parameters for D-CAL and X-CAL, we used $\gamma = 10000$ and the collection $\mathcal{C}$ was the set of 20 equally sized bins disjointed over $[0, 1]$.

### 5.2  RESULTS

**Average predicted survival functions without calibration.**  We compared the average survival functions predicted by the state-of-the-art neural networks with the Kaplan-Meier survival curve on each dataset. In our experiments, we used the fixed neural network and the loss functions presented in DeepHit (Lee et al., 2018), DRSA (Ren et al., 2019), and Survival CRPS (Avati et al., 2019). We used $\alpha = \sigma = 1.0$ for DeepHit and $\alpha = 0.25$ for DRSA as the hyperparameters. Figure 5 shows the average survival functions predicted by using these three loss functions and the Kaplan-Meier survival curve on each dataset. The average of the predicted survival functions should be close to the Kaplan-Meier survival curve if a prediction model is calibrated. However, none of the predicted results was close to the Kaplan-Meier survival curve, which means the loss functions do not achieve value-based calibration.

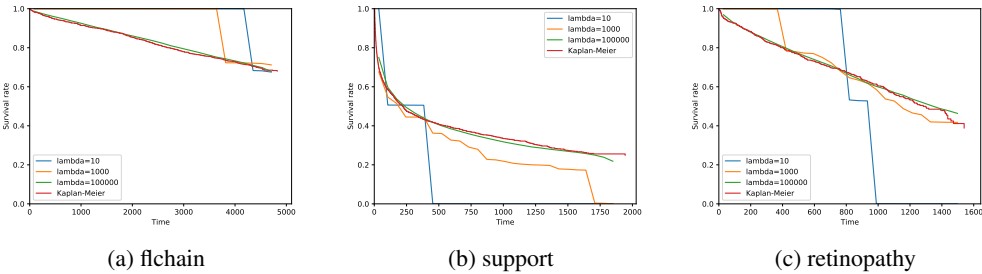

(a) flchain          (b) support          (c) retinopathy

Figure 6: Average predicted survival curves with Kaplan-Meier regularizer with varying $\lambda$.

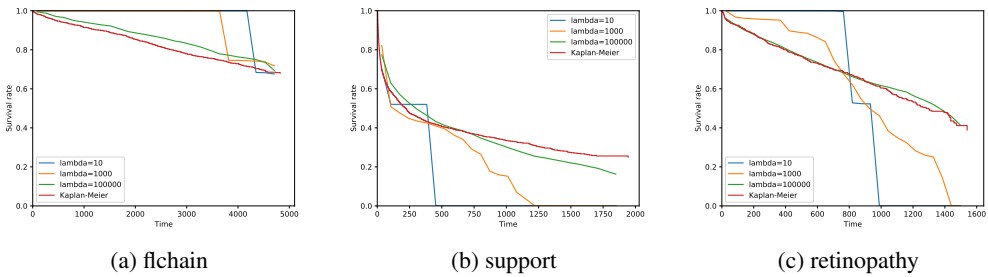

(a) flchain          (b) support          (c) retinopathy

Figure 7: Average predicted survival curves with X-CAL regularizer with varying $\lambda$.

**Calibration with our Kaplan-Meier regularizer and X-CAL.** We compared the performances of the proposed Kaplan-Meier regularizer and X-CAL by using each of them as a regularization term in conjunction with Survival CRPS (Avati et al., 2019). More specifically, we used the following loss function for the Kaplan-Meier regularizer:

$$\ell(\theta) = \ell_{\mathrm{S-CRPS}}(\theta) + \lambda \mathcal{R}_{\mathrm{KM}}(\theta),$$

where $\lambda$ is a parameter, and we used the following loss function for X-CAL:

$$\ell(\theta) = \ell_{\mathrm{S-CRPS}}(\theta) + \lambda \mathcal{R}_{\mathrm{X-CAL}}(\theta),$$

where $\lambda$ is the parameter. Figures 6 and 7 show results by using these loss functions with various $\lambda$ from $\{10, 10^3, 10^5\}$. We can see that the average predicted survival curves are not close to the Kaplan-Meier survival curve with small $\lambda$ and therefore the predictions were not calibrated. However, regarding the average predicted survival curves with $\lambda = 10^5$, the predicted curves with our Kaplan-Meier regularizer were close to the Kaplan-Meier survival curves and they are calibrated enough. In contrast, the predicted curves with X-CAL were close to the Kaplan-Meier survival curves, but the curves predicted by using our Kaplan-Meier regularizer were better than those with X-CAL.

Table 1 shows the D-CAL and the Kaplan-Meier loss of these predicted results. We can see here that using the Kaplan-Meier regularizer with large $\lambda$ leads to not only reducing the Kaplan-Meier loss but also reducing D-CAL. Similarly, using X-CAL with large $\lambda$ also leads to not only reducing D-CAL but also reducing Kaplan-Meier loss. These facts demonstrate that we can use one of these two regularizers (the Kaplan-Meier regularizer or X-CAL) to reduce the two metrics (Kaplan-Meier loss and D-CAL) for the real datasets, although we theoretically show that a quantile-calibrated model is not necessarily a value-calibrated model and vice versa in the appendix (Section A.1).

## 6 RELATED WORK

We use the Kaplan-Meier estimator (Kaplan & Meier, 1958) in our Kaplan-Meier loss to estimate the survival rate $S(t) = \mathrm{Pr}_{(\boldsymbol{X}, \boldsymbol{T}, \boldsymbol{C}) \sim Q}(\boldsymbol{T} > t)$. Our loss function is not restricted to the Kaplan-Meier estimator and we can use any other nonparametric method to estimate the survival rate $S(t)$. For example, we can use the Nelson-Aalen estimator (Aalen, 1978; Nelson, 1969; 1972) instead.

Table 1: D-CAL and Kaplan-Meier loss for the various combinations of dataset, regularizer, and $\lambda$

| | **Kaplan-Meier regularizer** (proposed) | $\lambda = 10$ | $\lambda = 10^3$ | $\lambda = 10^5$ |
|---|---|---|---|---|
| | D-CAL | 0.1181 | 0.0587 | 0.0023 |
| | Kaplan-Meier loss | 0.1110 | 0.0705 | 0.0004 |
| flchain | **X-CAL** (Goldstein et al., 2020) | $\lambda = 10$ | $\lambda = 10^3$ | $\lambda = 10^5$ |
| | D-CAL | 0.1176 | 0.0588 | 0.0005 |
| | Kaplan-Meier loss | 0.1110 | 0.0730 | 0.0049 |
| | **Kaplan-Meier regularizer** (proposed) | $\lambda = 10$ | $\lambda = 10^3$ | $\lambda = 10^5$ |
| | D-CAL | 0.2948 | 0.0938 | 0.0089 |
| | Kaplan-Meier loss | 0.0914 | 0.0167 | 0.0009 |
| support | **X-CAL** (Goldstein et al., 2020) | $\lambda = 10$ | $\lambda = 10^3$ | $\lambda = 10^5$ |
| | D-CAL | 0.0839 | 0.0186 | 0.0008 |
| | Kaplan-Meier loss | 0.0889 | 0.0426 | 0.0030 |
| | **Kaplan-Meier regularizer** (proposed) | $\lambda = 10$ | $\lambda = 10^3$ | $\lambda = 10^5$ |
| | D-CAL | 0.0800 | 0.0505 | 0.0010 |
| | Kaplan-Meier loss | 0.1731 | 0.0074 | 0.0004 |
| retinopathy | **X-CAL** (Goldstein et al., 2020) | $\lambda = 10$ | $\lambda = 10^3$ | $\lambda = 10^5$ |
| | D-CAL | 0.0783 | 0.0218 | 0.0003 |
| | Kaplan-Meier loss | 0.1733 | 0.0524 | 0.0005 |

Regarding another calibration for survival analysis, there was study on *1-Calibration* (Haider et al., 2020). In 1-Calibration, we consider the calibration of a single point of the predicted distribution, but the entire distribution is not evaluated for calibration. We also note that the distributional divergence for calibration (DDC) Kamran & Wiens (2021) is similar to D-CAL. The difference between DDC and D-CAL are that D-CAL uses a binning approach and DDC uses the Kullback Leibler divergence to measure the distance between the predicted survival function and the uniform distribution.

## 7 CONCLUSION

In this work, we presented new definitions of calibration, value-based calibration, for distribution regression and survival analysis. On the basis of the new calibration definition for survival analysis, we then proposed a new metric called Kaplan-Meier loss for value-based calibration. The results of experiments showed that it can be used as the regularizer of a loss function in a neural network model for calibration, and it can be seen as a simpler alternative to X-CAL.

## 8 REPRODUCIBILITY STATEMENT

We summarize our efforts for reproducibility. The details of our Kaplan-Meier loss and regularizer are described in Section 4. In addition, we describe the algorithm of the Kaplan-Meier estimator in Section 1 to make our paper self-contained, although the Kaplan-Meier estimator is a famous algorithm in survival analysis. Regarding our experiments, we described all of the details about our Python environment and the parameters used in Section 5. We used three datasets, and two of them are publicly available datasets. Although we do not attach our source code, we believe that readers of our paper can easily reproduce our results, because our algorithm is easy to implement and influence of the undescribed factors (e.g., choice of the random seed) can be ignored (e.g., the conclusion of our paper does not change even if the numbers in Table 1 fluctuate by 20%).

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

# A    APPENDIX

## A.1    RELATIONSHIP BETWEEN TWO DEFINITIONS OF CALIBRATION

We show that Definition 3.1 and Definition 4.1 are not equivalent. To see this, suppose that we have a probability distribution $\mathcal{P}$ on $\mathcal{X} \times \mathcal{Y}$, where $\mathcal{X} = \mathcal{Y} = [0, 1]$, such that the corresponding random variables $\boldsymbol{X}$ and $\boldsymbol{Y}$ are the uniform random variables on $[0, 1]$ that satisfy $\boldsymbol{X} = \boldsymbol{Y}$.

**A quantile-calibrated model is not a value-calibrated model.**    We consider a prediction model $F(y|x)$ defined by

$$F(y|x) = \left\{ \begin{array}{ll} 0 & (x = 0), \\ y & (0 < x < 1), \\ 1 & (x = 1). \end{array} \right.$$

Since the distribution $F(\boldsymbol{Y}|\boldsymbol{X})$ is the uniform distribution on $[0, 1]$, this prediction model $F(y|x)$ is quantile-calibrated by Definition 3.1. However, since we have

$$\mathbb{E}_{(\boldsymbol{X},\boldsymbol{Y}) \sim \mathcal{P}} F(y|\boldsymbol{X}) = 0.5$$

and

$$\Pr_{(\boldsymbol{X},\boldsymbol{Y}) \sim \mathcal{P}}(\boldsymbol{Y} \leq y) = y$$

for any $y \in (0, 1)$, the prediction model $F(y|x)$ does not satisfy the condition of calibration in Definition 4.1.

**A value-calibrated model is not a quantile-calibrated model.**    We consider a prediction model $F(y|x)$ defined by

$$F(y|x) = \left\{ \begin{array}{ll} 0 & (x > y), \\ 1 & (x \leq y). \end{array} \right.$$

Then this prediction model $F(y|x)$ is value-calibrated by Definition 4.1, because we have

$$\mathbb{E}_{(\boldsymbol{X},\boldsymbol{Y}) \sim \mathcal{P}} F(y|\boldsymbol{X}) = y = \Pr_{(\boldsymbol{X},\boldsymbol{Y}) \sim \mathcal{P}}(\boldsymbol{Y} \leq y)$$

for any $y \in [0, 1]$. However, since $F(y|x) = 1$ for any sample $(x, y) \sim \mathcal{P}$, the prediction model $F(y|x)$ does not satisfy the condition of calibration in Definition 3.1.

