# OpenReview forum: "Simpler Calibration for Survival Analysis"
_ICLR.cc/2022/Conference — ICLR 2022 Submitted_

### Official Review · Reviewer_KZxz · 2021-10-27

**Correctness:** 4
**Technical Novelty And Significance:** 2
**Empirical Novelty And Significance:** 2
**Recommendation:** 5
**Confidence:** 5

**Main Review:**

I found this paper enjoyable to read and the proposed method to be extremely intuitive and well-presented. Essentially the regularizer proposed is taking advantage of a consistency check, i.e., a reconstructed version of KM based on a user-specified survival model should match up with the classic KM estimator. The proposed KM regularizer is much simpler to describe than X-CAL and one doesn't have to deal with the clunky binning or censorship handling issues in the case of X-CAL. As an added benefit (one that perhaps is under-emphasized in the paper), the KM estimator is theoretically very well-understood with known rates of weak & strong convergence (e.g., Foldes & Rejto 1981a/b, Gill 1983, Chen & Lo 1997), known confidence interval construction (e.g., standard Greenwood formula). Of course, KM is also widely used/well-understood by practitioners.

I think this paper's main weakness is in the thoroughness of the experiments:
- It would be helpful to provide insights about various standard survival models that are used (beyond just picking a few SOTA approaches). For example, with the addition of the KM regularizer to different models, do we get some insight on which models tend to be better calibrated? Here, I think trying more models would be helpful. I realize that some thought would need to go into how to make the comparison in some sense fair across the models as different models' loss functions could have scales that are a bit different, so the same value of $\lambda$ across models doesn't correspond to the same regularization amount across models. Basically, the main question is whether some models just really struggle to be calibrated and one must use a very large amount of KM regularization, vs whether some models are just better-calibrated to begin with and very little if any KM regularization is needed.
- I think it would be helpful providing a baseline comparison against, say, random survival forests where the estimators at the leaves are KM estimators (rather than Nelson-Aalen estimators as in the original RSF paper), where you don't try to regularize RSF but you just show how far it is from KM. Similarly, seeing how close the standard Cox model's averaged individual survival curves compares with KM could also be helpful as a baseline comparison.
- If the survival model that we use is just the KM estimator so that we ignore input feature vectors entirely, then it would be perfectly calibrated according to the KM loss (if I'm understanding things correctly). Of course, this is a bit unsatisfying as the KM estimator does not provide individual-level information. By increasing the amount of KM regularization, we're making individual survival curves closer to the population KM average. This could mean that individual-level predictions are no longer as good. It would be helpful perhaps showing the tradeoff between a standard accuracy metric (e.g., time-dependent concordance or time-dependent AUC) vs the amount of KM/X-CAL regularization?
- I'd prefer having some sort of experimental repeats to get error bars of sorts on the experimental results (e.g., Figures 5-7, Table 1).

References:
- Kani Chen, Shaw-Hwa Lo. On the rate of uniform convergence of the product-limit estimator: strong and weak laws. The Annals of Statistics 1997.
- Antonia Foldes, Lidia Rejto. Strong uniform consistency for nonparametric survival curve estimators from randomly censored data. The Annals of Statistics 1981a.
- Antonia Foldes, Lidia Rejto. A LIL type result for the product limit estimator. Zeitschrift für Wahrscheinlichkeitstheorie und Verwandte Gebiete 1981b.
- Richard Gill. Large Sample Behaviour of the Product-Limit Estimator on the Whole Line. The Annals of Statistics 1983.

**Summary Of The Paper:**

This paper proposes a new calibration loss motivated by a simple observation: the average of individual estimated survival curves should be close to the marginal nonparametric estimate given by the Kaplan-Meier estimator. This calibration loss readily supports mini-batching, introduces no new hyperparameters during the optimization, and is shown to also be effective at encouraging an existing calibration metric D-CAL to be small.

**Summary Of The Review:**

The proposed KM regularizer looks very promising and is quite intuitive. While the experiments presented so far are instructive, they could be substantially more thorough to help us understand the pros vs cons of using more KM regularization, and also what we can say about how well-calibrated different standard survival models are.

---

### Official Review · Reviewer_MTaz · 2021-10-29

**Correctness:** 3
**Technical Novelty And Significance:** 3
**Empirical Novelty And Significance:** 2
**Recommendation:** 5
**Confidence:** 3

**Main Review:**

=== Strengths

One of the main strengths of this work is the novelty, yet relative simplicity, of the proposed approach. Defining calibration of the learned survival function with respect to the actual Kaplan-Meier (or similar) curve is reasonable. I am not an expert in this area, but to the best of my knowledge, it is also novel.

I did not thoroughly verify the propositions and assertions of probability, though they seem to follow from definitions. Thus, the derivation of the proposed regularization terms appears sound.

=== Weaknesses

The empirical evaluation comparing the proposed and existing approaches is rather limited. First, the dataset characteristics which result in performance differences of the two approaches is not explored at all. For example, on the “flchain” dataset, the proposed approach results in an order of magnitude improvement for the KM-loss compared to X-CAL. On the other hand, for “retinopathy,” there is hardly any difference. It would be helpful to understand the characteristics of these datasets (such as amount of censoring, spread of survival times, types or amount of features, etc.) which lead to different relative performance. If needed, synthetic datasets could be created to control various factors of the data, such as the rate of censoring. Second, it would be very useful to also report some measures of prediction accuracy; as the authors highlight, both calibration and sharpness are important. Third, cross validation or some other approaches should be used to provide some measure of statistical significance for the work.

The clarity of the paper could be improved. While the overall structure and flow of the paper is reasonable, Sections 3 and 4 are quite dense. As a reader less familiar with this area, I would have prefered to avoid the various reformulations in the middle of the text; rather, I believe starting with and sticking with the forms (possibly refering to the equivalences given in Section 2) finally used in the regularizers would have been easier to follow.

The reproducibility and value of resources provided by the authors is clearly limited. The proposed approach itself seems straightforward enough to implement, as the authors say; however, considering the numerous steps in the method, the lack of reproducibility diminishes the impact of the work.

=== Typos, etc.

The font size in Figures 5, 6, and 7 should be increased.

“for the real datasets” -> “for real datasets”

In the [Kamran and Wiens] reference: “In AAAI” -> “In Proceedings of AAAI”

**Summary Of The Paper:**

In this work, the authors propose a novel approach for learning calibrated predictions for survival analysis and similar tasks. The intuition of the approach is that the average probability that a prediction has a value less than or equal to $t$ should approximately equal the number of observations with value less than or equal to $t$. The authors then derive a differentiable regularization term using this idea and Kaplan-Meier curves. A small set of empirical evaluations suggest the proposed approach modestly outperforms another recent survival analysis calibration method according to some metrics.

**Summary Of The Review:**

Time-to-event predictions accounting for censored data is a wide-spread problem. Considering the relative simplicity, such as lack of hyperparameters, of the proposed approach, I believe it has the potential to have modest impact in several applied areas. On the other hand, the empirical support for the work is rather limited, so more evidence is likely needed before wider adoption and impact.

---

### Official Review · Reviewer_sM7d · 2021-11-02

**Correctness:** 3
**Technical Novelty And Significance:** 1
**Empirical Novelty And Significance:** 1
**Recommendation:** 3
**Confidence:** 4

**Main Review:**

The paper explores calibration in survival analysis, which is an important yet relatively under-explored problem. However, I found several issues with this submission. Here is a summary:

*The paper overlooks several works in survival analysis including an important baseline*
- A Kaplan-Meier calibration objective for survival analysis has been proposed before [1]. Therefore the paper (including experimental results) should be positioned relative to [1]
- Prominent survival analysis literature before neural networks have been completely ignored, *e.g.*, cox proportional hazards [2], accelerated failure time times [3], *etc.*
- Several works (including neural network-based) that focus on hazards (instantaneous risk) predictions have been ignored, including [2, 4].
- While the paper references S-CRPS, the paper overlooks that the S-CRPS objective focuses on sharp and calibrated predictions. A discussion connecting their proposed objective to the S-CRPS thresholded calibration with Brier scores is important since Brier scores are prominently used as a calibration metric in survival analysis.

*Weak experimental evaluations*
- The paper's focus on only calibration metrics is concerning since calibration alone does not yield accurate predictions. Other accuracy-based metrics such as concordance index and relative absolute error should be included. This is important since calibration and accuracy are orthogonal concerns.
- As expected, it seems the Kaplan-Meier regulariser and X-CAL obtain perfect calibration at very large $\lambda$. At such a large $\lambda$, the model survival predictions will be inaccurate. Therefore, the paper should provide sensitivity analysis on how accuracy is impacted at varying $\lambda$.
- Why is the empirical Kaplan-Meier compared to estimated average survival functions, instead of an estimated Kaplan-Meier from predictions, since the datasets are right-censored?
- Since X-CAL has a slight competitive advantage. Why should users adopt the proposed Kaplan-Meier regulariser?

*Misleading statement*
-  Abstract: "While there have been many neural network models proposed for survival analysis, none of them are calibrated." This is a very strong and misleading statement. Depending on the calibration objective, some methods are calibrated, *e.g.*, S-CRPS, SFM in [1], *etc.*

*Redundant content*
- In general, writing can be improved by minimizing the discussions focused on *regression* as they may serve as a distraction since survival analysis is completely different from regression.

*Minor Issues*
- There are several typos, I encourage the authors to proofread carefully

**Missing references**
- [1] Chapfuwa et al., "Calibration and Uncertainty in Neural Time-to-Event Modeling", IEEE Transactions on Neural Networks and Learning Systems, 2020.
- [2] Cox, “Regression models and life-tables,”  Breakthroughs in Statistics, 1992.
- [3]  Wei, “The accelerated failure time model: A useful alternative to the cox regression model in survival analysis,” Statistics in medicine, 1992
- [4] Katzman et al., "Deep survival: A deep cox proportional hazards network", BMC Medical Research Methodology, 2018

**Summary Of The Paper:**

The paper proposes a Kaplan-Meier calibration regulariser for training survival (right-censored) datasets for calibrated predictions. Experimental results of the proposed approach against previously proposed  X-CAL regulariser on three datasets per metrics: D-CAL and Kaplan-Meier loss indicate that X-CAL has a competitive advantage.

**Summary Of The Review:**

The paper overlooks several prominent works in survival analysis, including a baseline closely related to their proposed Kaplan-Meier regulariser. Additionally, the paper's focus on calibration alone in the experimental results is problematic since calibration and accuracy are equally important but orthogonal metrics.

---

### Official Review · Reviewer_xdbC · 2021-11-04

**Correctness:** 3
**Technical Novelty And Significance:** 2
**Empirical Novelty And Significance:** 2
**Recommendation:** 3
**Confidence:** 4

**Main Review:**

Title:
    > "simpler" compared to what? Also, what does "simple" mean in this context? "easier to optimize"?


Abstract:
    > In your abstract and even at the beginning of the introduction, it is not obvious what you are trying to estimate. Survival function? hazard? What is your neural network trying to predict?
    > You put much more emphasis on X-CAL rather than your own work. I believe it would have been better to talk about your work in this paper.


Section 1:
    > first paragraph: you talk about survival analysis, then talk about application in healthcare, then again you talk about survival analysis and after than about its application in healthcare and other fields. The flow fluctuates. Also, you just talk about the existence of a survey paper without mentioning what to take away from it.
    > third paragraph: The connection between KM and your work at this point is not properly explained given you are delving into it. After that, you only discuss other models that are patient-specific. But the connection to the main paper is still not clear. Your work, as far as I understood, is not making KM patient specific, it is about using it as a regularizer.


Section 1.1:
    > first paragraph: please define what you mean by confidence.


Section 1.2:
    > same as the title, 'simpler' in what sense?
    > "an advantage ... is calibrated for any time y \n [0, t_{max}]." why is this an advantage?


Section 2:
    > at the end, also a branch of work has been dedicated to hazard estimation rather than survival function estimation.


Section 3 and 4:
    > prior work seems to be mixed up with your work. It would have been more organized to first talk about prior work (calibration definition and KM and things like that) and then move to your own work. This way they can be mixed up. Or talk about your work first and then the connection to the prior work.


Section 3:
    > after equation 1, maybe worth mentioning this is due to the CDF function being nondecreasing and if invertible, the inequality does not flip
    > last paragraph: explaining the work already done in Goldstein et al. it should be more aligned towards your work and how it is different from the prior work, not explaining somebody else's work.


Section 4:
    > several definitions of calibration make it confusing which one is the one you are considering. Same goes for Section 3 vs Section 4.
    > at equation 8 and still not sure what your model is. Is your neural network predicting the parameters of a known distribution? It is not obvious to me how you compute equation 8 given the model is not known but you talk about mini-batch training which means your model should be defined concretely already.
    > The advantages of the approach: advantage 1: why should your regularization term or even the calibration be 'intuitive'? either your definition is well-calibrated, ready to be used, or not.


Section 5:
    > In the only paragraph for this section, it seems like the calibration definition is relative and there is no better way of defining it.
    > How about discrimination results? You are regularizing the model based on KM which is not input dependent. I am curious to see how it affects the model discrimination or 'sharpness' as mentioned in the paper.


Section 5.2:
    > You are including a KM based regularization in your model so the significance of average survival curve being similar to the KM curve needs to be explained. Also in the last paragraph, what is the take-away? Which of the two regularizers prove to be more useful?

**Summary Of The Paper:**

This paper first defines a new definition of calibration which, in contrast to the one used in prior work, is confined to the maximum observed time in the data. They then propose a KM regularizer for making sure their survival curves are calibrated: they are closer to KM curves. They also claim that their regularization, unlike prior work, does not require any hyperparameters and avoids the binning approach.

**Summary Of The Review:**

The paper needs organization and the significance needs to be more clear. Also, I am skeptical about how much better it performs over the baseline given the authors only give results on their defined calibration rather than how discriminative the model remains. When optimizing for curves similar to KM, it is expected to obtain them as such. How would the other metrics change?

---

### Decision · Program_Chairs · 2022-01-20

**Decision:**

Reject

**Comment:**

A number of suggestions have been given about the manuscript. The evaluation raised questions about clarify, placement with respect to other approaches, choices for the design, etc. There are no immediate replies from authors, so I hope the suggestions are useful for future work.